# A Coverage Hole Patching Algorithm for Heterogeneous Wireless Sensor Networks

**Xinmiao Lu [1],\*, Yuhan Wei [1], Qiong Wu [2], Cunfang Yang [1], Dongyuan Li [1], Liyu Zhang [1] and Ying Zhou [2]**

[1] School of Measurement-Control Technology and Communications Engineering, Harbin University of Science and Technology, Harbin 150080, China
[2] Heilongjiang Network Space Research Center, Harbin 150090, China
\* Correspondence: lvxinmiao0611@hrbust.edu.cn; Tel.: +86-187-0462-8841

**Abstract:** The improvement of coverage is a critical issue in the coverage hole patching of sensors. Traditionally, VOPR and VORCP algorithms improve the coverage of the detection area by improving the original VOR algorithm, but coverage hole patching algorithms only target homogeneous networks. In the real world, however, the nodes in the wireless sensor network (WSN) are often heterogeneous, i.e., the sensors have different sensing radii. The VORPH algorithm uses the VOR in a hybrid heterogeneous network and improves the original algorithm. The patched nodes are better utilized, and the detection range is enlarged. However, the utilization rate of the patched nodes is not optimized, making it impossible to patch the coverage holes to the maximum degree. In the environment of hybrid heterogeneous WSN, we propose a coverage hole patching algorithm with a priority mechanism. The algorithm determines the patching priority based on the size of the coverage holes, thereby improving network coverage, reducing node redundancy, and balancing resource allocation. The proposed algorithm was compared under the same environment by simulation and analysis. The results show that our algorithm is superior to the traditional coverage hole patching algorithms in coverage rate, and can reduce node redundancy.

**Keywords:** coverage hole; hybrid heterogeneous wireless sensor network (WSN); priority mechanism; patching

## 1. Introduction

Classic harmful gas detection mainly uses wired fixed devices and portable instruments and other detection methods with poor flexibility, low real-time, inaccurate leakage point positioning and other shortcomings. A wireless sensor network with low cost, high real-time, good cooperation and other characteristics to achieve remote detection of harmful gases is a new idea. However, the wireless sensor network is a resource-limited network. How to use limited sensor nodes to improve network coverage performance in the wireless sensor network in the application of harmful gas detection has technical difficulties.

The deployment of network nodes determines the coverage performance of the network [1]. In the natural environment, it is impossible to predetermine the specific situation of the WSN in many monitoring applications due to the inaccessibility of personnel and the complexity and randomness of the environment [2]. Therefore, the sensors are often deployed randomly, without pre-determining the location of nodes in the monitoring area. Random deployment may greatly reduce the WSN coverage of the monitoring area, leaving coverage holes [3]. What is worse, wireless sensor nodes have a limited lifetime, and when energy is exhausted, the node fails, creating a coverage hole. Wireless sensors may fail due to various reasons, such as high temperature, malicious attacks, harsh environments, overuse, and lack of energy. Coverage holes appear right after the failure [4].

Once generated, the coverage holes affect the service quality of the entire network, making the network unable to capture the data of the object accurately, or to realize the intended purpose [5,6]. Network coverage, an important indicator of the performance of WSN, depicts

the quality of network service and network perception ability. The appearance of coverage holes significantly squeezes the coverage of WSN, undermining the coverage performance [7].

Yu et al. [8] improved the Voronoi-based optimization algorithm (VOR) into two coverage hole patching algorithms, namely, VORP and VORCP. The improvement enhances the coverage of the test area. Zhou et al. [9] proposed an improved coverage hole patching enhancement algorithm based on virtual force relocation. The improved algorithm adopts the Delaunay triangle algorithm to detect the local coverage holes between sensors, and then adopts the virtual force algorithm to patch the holes. The algorithm speeds up convergence, while elevating the network coverage. Cao et al. [10] devised the FISS algorithm for selecting patching nodes based on the fuzzy inference system (FIS). First, the coverage holes were identified by the adaptive sensing radius adjustment strategy, before patching the coverage holes. The FISS algorithm not only effectively improves the coverage rate, but also controls the energy consumption. Hao et al. [11] developed a three-dimensional (3D) dynamic detection and patching algorithm for coverage holes. The algorithm improves the node utilization rate and lowers the network coverage cost. It can meet the overall requirement of network coverage with fewer nodes, and consumes less mobile energy. The above algorithms work well in homogeneous networks, but the WSN nodes in the actual environment are often heterogeneous. In other words, the sensors in a heterogeneous network differ in sensing range, which results in coverage holes. Thus, the above algorithms are not effective in patching coverage holes in heterogeneous networks. Wu et al. [12] presented a coverage hole patching algorithm called VORPH, which uses the VOR in a hybrid heterogeneous network and improves the original algorithm. The patched nodes are better utilized, and the detection range is enlarged. However, the utilization rate of the patched nodes is not optimized, making it impossible to patch the coverage holes to the maximum degree. Each algorithm has limitations [13–15].

To solve the defects of the above algorithms in heterogeneous networks, we propose a coverage hole patching algorithm with priority mechanism in the environment of hybrid heterogeneous WSNs. Using the hybrid heterogeneous WSN, the we fixed static sensors to the area to be tested randomly, creating coverage holes for various reasons. Next, the sensors were moved by the priority mechanism to patch the coverage holes of the hybrid heterogeneous network.

## 2. Network Model and Coverage Hole Positioning Algorithm

### 2.1. Network Model

Suppose the rectangular detection area A is divided into m × n grids. A total of N static sensors $S = \{S_1, S_2 \cdots S_N\}$ are randomly deployed in the area. The static nodes fall into two categories: the nodes with a large sensing radius $R_{S1}$, and those with a small sensing radius $R_{S2}$. The radius of the former nodes is 1.5 times that of the latter nodes. The latter nodes are selected as the movable sensors that patch the coverage holes. The communication radii of the nodes with a large sensing radius and those with a small sensing radius are denoted by $R_{C1}$ and $R_{C2}$, respectively, and $R_{C1} = 2R_{S1}$, $R_{C2} = 2R_{S2}$, and $R_{S2} = 1.5R_{S1}$. The communication radius length $R_C$ between sensor nodes is twice the sensing radius $R_S$, ensuring that sensors can communicate with each other. The communication graphs between the sensors are connected. Each sensor can obtain the information (e.g., position) of other sensors.

In the real environment, the sensors with a large sensing radius will not surpass those with a small sensing radius by 0.5 times in terms of the sensing radius. The reason is that the overall cost of the system will hike, if sensors have an excessively large sensing radius. Let the center $S_i = (x_i, y_i)$ of a circle be the coordinates of a node. The area within the circle is the coverage of the WSN. The mobile nodes belong to M categories. The number of categories is the same with that of the sensors with a sensing radius of $R_{S1}$. In WSN research, network coverage and redundancy are two important metrics, which can be calculated by the following formulas.

1. Perceptibility

The perceptibility of node $S_i$ for target $t$ can be calculated by:

$$p(s_i, t_j) = \begin{cases} 1, & d(s_i, t_j) \leq R_s \\ 0, & d(s_i, t_j) > R_s \end{cases} \tag{1}$$

where $d(s_i, t_j)$ is the distance from the circle center of the sensor to the target. If target $t$ is covered, the perceptibility of node $S_i$ for target t equals 1; otherwise, the perceptibility is 0.

2. Network coverage

Coverage, the most crucial indicator of WSN performance, can be calculated by:

$$R_{cov}(s) = \frac{A_{cov}(s)}{A_{total}} = \frac{\sum_{x=1}^{m} \sum_{y=1}^{n} p(x, y, s)}{m \times n} \tag{2}$$

where $A_{cov}$ is the coverage area, $A_{total}$ is the area of the detection area, m is the length of the detection area and $n$ is the width of the detection area.

3. Network redundancy

Redundancy indicates the degree of overlap between sensors. The greater the redundancy, the less even the distribution of the sensors. The network redundancy can be calculated by:

$$R_r(s) = \frac{A_r}{A_{cov}} = \frac{\sum_{x=1}^{m} \sum_{y=1}^{n} p_r(x, y, s)}{\sum_{x=1}^{m} \sum_{y=1}^{n} p(x, y, s)} \tag{3}$$

where $A_{cov}$ is the area of the sensor coverage, $A_r$ is the overlapping area between sensors, and $p_r(x, y, s)$ indicates that grid node $(x, y)$ can be perceived by sensor $S_i$ and sensor $S_j$.

*2.2. Covering Hole Positioning Based on Circumscribed Circles*

In Figure 1, circles A, B, and C are the initially deployed static nodes, among which a hole is generated. Make a circle O that circumscribes circles A, B, and C at the same time. The larger the area of circle O, the wider the coverage hole, and the greater the coverage propriety. When the radius of circle O is less than 1/4 of the radius of the sensors with small sensing range, it is not recorded as a coverage hole, and the center O of circle O is the location where the mobile nodes will move to.

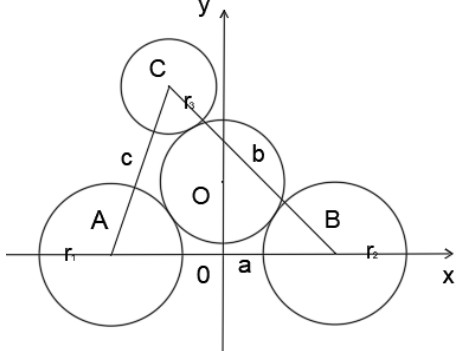

**Figure 1.** Determining the specific location of the node to be patched.

In Figure 1, the radii of the three circles A, B, and C on the known plane are denoted by $r_1$, $r_2$, and $r_3$, respectively, with $r_1 = r_2 = 1.5r_3$; AB $= a$, BC $= b$, and CA $= c$. Taking the straight line of segment AB as the horizontal axis, and the midpoint of AB as the origin, a rectangular coordinate system is established, with point C falling on the upper half plane of the rectangular coordinate system. Suppose A$(-a/2, 0)$, B$(a/2, 0)$, and C$(x_3, y_3)$. Let

$O(x, y)$ and $r$ be the center and radius of the circle O to be solved, respectively. Then, C can be determined by the following equation set (note that $y_3 > 0$):

$$\begin{cases} (x_3 - \frac{a}{2})^2 + y_3{}^2 = b^2 \\ (x_3 + \frac{a}{2})^2 + y_3{}^2 = c^2 \end{cases} \Rightarrow \begin{cases} x_3 = \frac{c^2 - b^2}{2a} \\ y_3 = \frac{2S}{a} \end{cases} \tag{4}$$

where $p = \frac{a+b+c}{2}$ is the half perimeter of $\Delta_{ABC}$. The area of the circle can be described by Heron's formula as $S = \sqrt{p(p-a)(p-b)(p-c)}$. Then, the center $O(x, y)$ of circle O can be determined by the following equation set:

$$\begin{cases} (x + \frac{a}{2})^2 + y^2 = (r + r_1)^2 \\ (x - \frac{a}{2})^2 + y^2 = (r + r_2)^2 \\ (x + x_3)^2 + (y - y_3)^2 = (r + r_3)^2 \end{cases} \tag{5}$$

Thus, we have:

$$\begin{cases} x = \frac{(r_1 - r_2)(2r + r_1 + r_2)}{2a_3} \\ y = \frac{\sqrt{-[a^2 - (r_1 - r_2)^2][a^2 - (2r + r_1 + r_2)^2]}}{2a} \end{cases} \tag{6}$$

Substituting $O(x, y)$ into Formula (5), the same solution $r = \frac{P \pm 4S \cdot Q}{R}$ can be obtained, with $P$, $Q$ and $R$ being:

$$P = \sum_{cyc} b^2 [-b^2 r_1 + c^2 (r_1 + r_2) - (r_1 - r_2)(r_1 - r_3)(2r_1 + r_2 + r_3)] \tag{7}$$

$$Q = \prod_{cyc} \sqrt{b^2 - (r_2 - r_3)^2} \tag{8}$$

$$R = \sum_{cyc} b^2 [b^2 - 2c^2 + 4(r_1 - r_2)(r_1 - r_3)] \tag{9}$$

In this way, the location of the circle O to be solved can be obtained, that is, where the mobile nodes to be moved to.

## 3. Improved Coverage Hole Patching Algorithm with Priority Mechanism

We propose a coverage hole patching algorithm for hybrid heterogeneous WSN with priority mechanism based on circumscribed circles. The priority mechanism assigns a priority to each hole to be patched. First, the holes are compared in size, and the larger hole is given a higher priority. Then, a limited number of mobile nodes is used to patch the holes by the priority, thereby optimizing the coverage.

### 3.1. Situations of Coverage Holes

In Figure 2, N is the center of the circle. The nodes A, B and C are far apart, resulting in very large coverage holes. A mobile node is set at the center of the common circumscribed circle of the three nodes. Although it does not overlap nodes A, B and C, the mobile node patches the coverage holes to the maximum possible extent. In addition, since the distance between nodes A, B and C exceeds the communication distance, the mobile node increases the communication coverage of nodes A, B and C, and enhances the connectivity of communication.

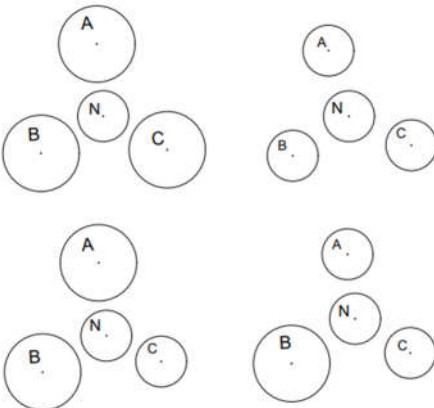

**Figure 2.** Nodes A, B and C are far from each other.

In Figure 3, the distance between nodes A, B and C is relatively short, resulting in a large coverage hole. A mobile node is set at the center of the common circumscribed circle of the three nodes to cover nodes A, B and C, thereby making up the hole.

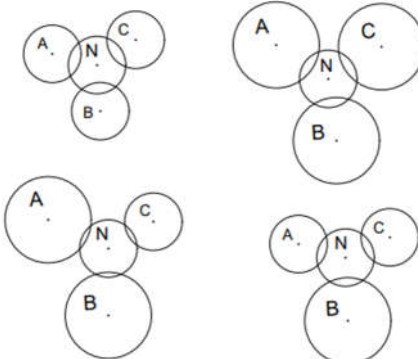

**Figure 3.** Nodes A, B and C are close to each other.

In Figure 4, two nodes A and B intersect each other, and the other node C has no overlap with A or B. The sensing ranges of nodes A and B cover each other, resulting in a large coverage hole. A mobile node is set at the center of the common circumscribed circle of the three nodes so that the position of the mobile node can maximize the patching of the hole.

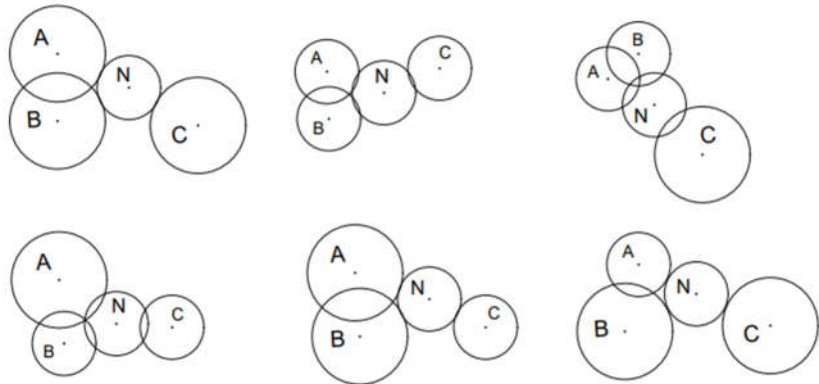

**Figure 4.** Nodes A and B intersect each other.

In Figure 5, nodes A and B intersect, and nodes B and C intersect, but A and C do not intersect. The sensing ranges of nodes A and B cover each other, and the sensing ranges of nodes B and C cover each other. In the common circumscribed circle of the three nodes, a mobile node is set at the center of the circle to cover nodes A, B and C to repair the hole. The setting of the position of the mobile node maximizes the patching of the hole.

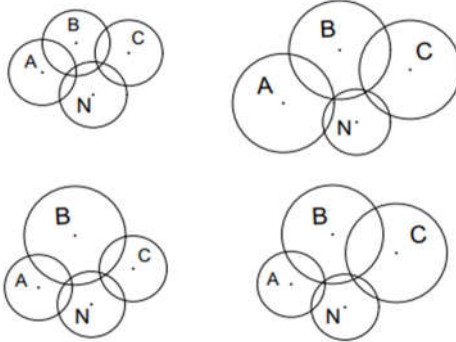

**Figure 5.** Multiple nodes intersect each other.

As shown in Figures 2–6, the hole in Figure 2 is the largest, and its patching priority is the highest. For Figures 3–5, the patching priority increases with the size of the circumscribed circle. Figure 6 needs not to be covered, for no hole is generated.

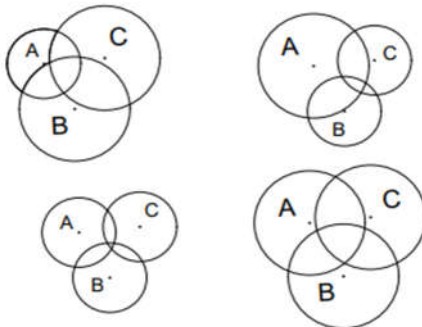

**Figure 6.** Nodes A, B and C intersect each other in pairs.

*3.2. Steps*

The specific hole patching steps are as follows:

1. After randomly arranging static nodes, obtain the specific location of each node and store it in the list of locations v.
2. Create a new list of the patching locations List and a temporary information list temp.
3. Divide the adjacent three nodes into a group, make a common circumscribed circle of each group of three nodes, calculate the area of the circumscribed circle $G_1 \cdots G_N$, and store it into the List.
4. Initialize the temporary information list temp.
5. Carry out a loop, traverse the list, directly update the first $G_1$ in the list and its related location information into temp. Continue to traverse the list, and take $G_2$ to compare it with $G_1$ in temp. If $G_2 > G_1$, add $G_2$ and its related location information into temp, and remove $G_1$ and its related location information from the list. Otherwise, retain $G_1$ in temp. When all items in the list are traversed, send the marked locations in the List to the mobile node to guide its movement.
6. Repeat Steps 3–5 until the mobile nodes are all used, and terminate the loop.

## 4. Simulation Experiments

To demonstrate the effectiveness of our algorithm, network coverage and redundancy were analyzed, using the software simulation platforms of Win10 and Matlab 2016b.

The detection area is of the size 125 m × 100 m. A total of 25 static nodes were deployed randomly inside the area, $S = \{S_1, S_2 \cdots S_{25}\}$. The nodes fall into two categories. The sensing radii of the two types of nodes are $R_{S1}$ and $R_{S2}$, respectively; the communication radii of the two types of nodes are $R_{C1}$ and $R_{C2}$, respectively. These radii satisfy $R_{C1} = 2R_{S1}$, $R_{C2} = 2R_{S2}$, and $R_{S2} = 1.5R_{S1}$. The sensing radii of the nodes are $R_{S1} = 8$ m and

$R_{S2} = 12$ m. The communication radii assure that the nodes within the one-hop range can exchange information. The center $S_i = (x_i, y_i)$ of each circle represents the coordinates of a node. The area enclosed in the circle indicates the coverage of the WSN. The types and sensing radii of the mobile nodes are the same as those of the sensors with a sensing radius of $R_{S1}$. There are a total of 15 sensors. Simulation experiments were carried out to verify the proposed method. Figure 7 shows the coverage of the static nodes. The coverage was calculated to be 62.46%. There are 27 potential coverage holes in the detection area, which is larger than the number of mobile nodes that can be used.

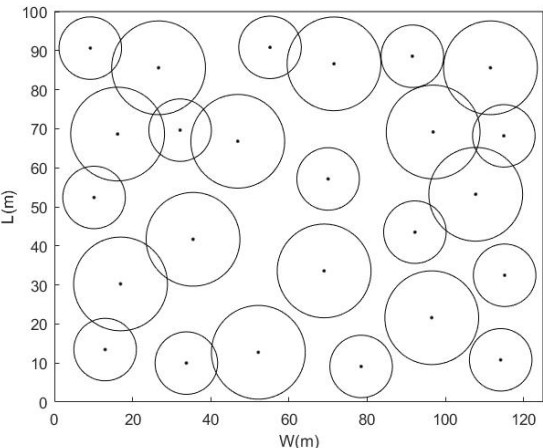

**Figure 7.** Randomly assigned static nodes.

The sizes of the coverage holes of the static nodes in Figure 7 are different. Our algorithm was adopted to simulate the patching of the coverage holes. The simulation results are shown in Figure 8. The calculated coverage rate was 79.13%, which is 16.67% higher than the coverage rate before patching. The redundancy was 8.24%. This means the coverage holes of the WSN have been well repaired, reflecting the effectiveness of our algorithm.

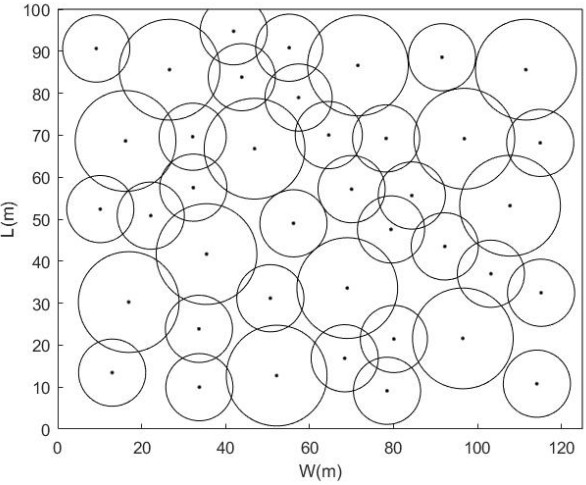

**Figure 8.** Nodes after patching by our algorithm.

Using the same parameters, our algorithm was compared with VORCP [1] and VORPH [2] in terms of coverage and redundancy, with the goal of revealing the superiority of our algorithm intuitively.

Figure 9 shows the simulation results on patching the coverage holes in the scene of Figure 7, using VORCP. The coverage (75.04%) was 12.58% greater than that before patching. The redundancy was 10.54%. Figure 10 shows the simulation results on patching the coverage holes in the scene of Figure 7, using VORPH. The coverage (76.78%) was 14.32% greater than that before patching. The redundancy was 11.13%. Table 1 shows

that our algorithm performed the best in hole patching and coverage enhancement, under the same environment. In addition, the redundancy after patching of our algorithm was lower than that of the other two algorithms. This fully demonstrates the superiority of our algorithm. Table 1 shows the comparison result of the algorithms.

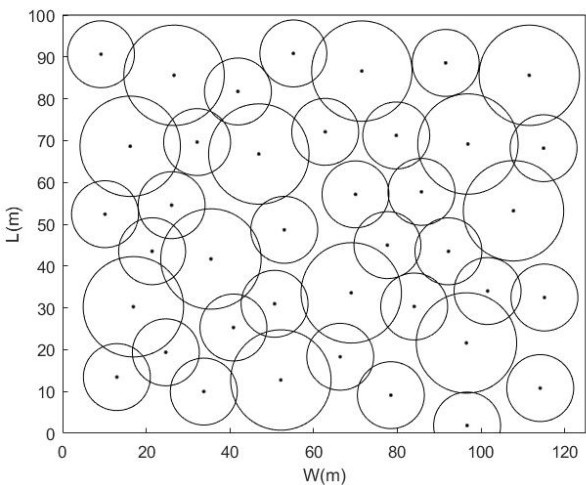

**Figure 9.** Nodes after VORCP patching.

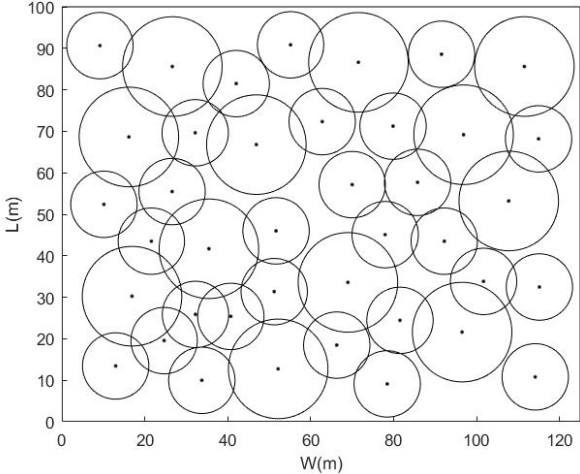

**Figure 10.** Nodes after VORPH patching.

**Table 1.** Algorithm comparison.

|  | Coverage | Increased Coverage | Redundancy |
|---|---|---|---|
| VORCP | 75.04% | 12.58% | 10.54% |
| VORPH | 76.78% | 14.32% | 11.13% |
| Our Algorithm | 79.13% | 16.67% | 8.24% |

## 5. Conclusions

To solve the defects of traditional coverage hole patching algorithms, we proposed a coverage hole patching algorithm with priority mechanism in hybrid heterogeneous WSN environment. Specifically, we analyzed the network model and the hole positioning algorithm, and studied the coverage and priority of the coverage holes. Then, our algorithm was subjected to simulation and analysis, and compared with VORCP and VORPH. The results show that, in the hybrid heterogeneous WSN, our algorithm outperformed the traditional coverage hole patching algorithms in both coverage and redundancy.

**Author Contributions:** Conceptualization, X.L.; methodology, X.L.; software, Y.W.; validation, X.L., Q.W. and D.L.; writing—original draft preparation, C.Y., L.Z. and Y.Z.; writing—review and editing, Y.W.; project administration, X.L.; funding acquisition, X.L. All authors have read and agreed to the published version of the manuscript.

**Funding:** This research was funded by Heilongjiang Provincial Natural Science Foundation of China, grant number YQ2022F014.

**Acknowledgments:** The authors acknowledge Heilongjiang Provincial Natural Science Foundation of China (grant number YQ2022F014).

**Conflicts of Interest:** The authors declare no conflict of interest.

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
