# Peer review of "A Coverage Hole Patching Algorithm for Heterogeneous Wireless Sensor Networks"

_electronics, doi:10.3390/electronics11213563_

Round 1
Reviewer 1 Report
The manuscript is well written, well presented and contains novel idea. Further the manuscript is correct both scientifically and grammatically. It will help a lot to researchers working in the direction of wireless sensors network. Hence I recommend it to be accepted for publication in its current form.
Author Response
Dear reviewer:
Thank you for your decision and constructive comments on my manuscript. We have carefully considered the suggestion of the Reviewer and made some changes. We have tried our best to improve it, polishing the language and making some changes to the manuscript.
Reviewer 2 Report
Dear Author(s)
The paper is containing a very good idea and the style of writing is well also. But the manuscript is need to moderate checking of the English language and little modified the conclusions and results.
Author Response

(The authors gave the same response as above.)

Reviewer 3 Report
This paper presents an interesting work. The author has emphasized an approach or algorithm to improve the hole patching issues in wireless sensor networks. However, need to consider the following comments.
1. Abstract was not adequate. Authors need to update by mentioning the limitations in the existing work and supposed to mention their work over them.
2. Introduction part may be extended by including information related to attacks and security flaws.
3. Literature survey may be extended by incorporating some explanation related to existing wireless sensor network applications.
4. What are the attacks that can be mitigated by this proposed method?
5. What is the session time or lifetime of the randomly generated static nodes? Are they permanent nodes or temporary nodes?
6. How routing is performed among the randomly assigned static nodes in WSN?
7. Mentioned Win10 and Matlab2016b were used here to get the simulation results. Needs to elaborate on the implementation specification of the proposed work.
8. How many nodes authors have considered while in implementation?
9. Authors mentioned the proposed algorithm compared with VORCP and VORPH. Add a table to show the comparison results with the existing systems.
10. Suggested consider the following works on WSN
a. 10.1109/WiSPNET51692.2021.9419459
b. https://doi.org/10.1007/978-81-322-2126-5_18
c. https://doi.org/10.1155/2022/9134036
.
Author Response
Dear reviewer:
Thank you for your decision and constructive comments on my manuscript. We have carefully considered the suggestion of the Reviewer and made some changes. We have tried our best to improve and made some changes to the manuscript.
The yellow part has been revised according to your comments in this paper. Revision notes, point-to-point, are given as follows:
Point 1: Abstract was not adequate. Authors need to update by mentioning the limitations in the existing work and supposed to mention their work over them.
Response 1:
We've added to the summary section, and here's my summary:
Traditionally, VOPR and VORCP algorithms improve the coverage of the detection area by improving the original VOR algorithm,but coverage hole patching algorithms only target homogeneous networks. In the real world, however, the nodes in the wireless sensor network (WSN) are often heterogeneous, i.e., the sensors have different sensing radii. VORPH algorithm uses the VOR in a hybrid heterogeneous network and improves the original algorithm. The patched nodes are better utilized, and the detection range is enlarged. However, the utilization rate of the patched nodes is not optimized, making it impossible to patch the coverage holes to the maximum degree.
Point 2: Introduction part may be extended by including information related to attacks and security flaws.
Response 2:
Thank you very much for your valuable comments, the research direction of this paper is to use the priority mechanism to patch the algorithm, improve the network detection coverage within the detection range, reduce node redundancy, and do not study the direction related to attacks and security vulnerabilities, in future research, we will increase the research on this direction.
Point 3: Literature survey may be extended by incorporating some explanation related to existing wireless sensor network applications.
Response 3:
The background of the research project of this paper is the research of harmful gas detection and data transmission technology in the wireless sensor network industry, and application-related explanations have been added to the paper.
The classic harmful gas detection mainly has wired fixed devices and portable instruments and other detection methods, there are poor flexibility, low real-time, inaccurate leakage point positioning and other shortcomings, wireless sensor network with its low cost, high real-time, good cooperation and other characteristics to achieve remote detection of harmful gases provides a new idea. However, a wireless sensor network is a resource-limited network, how to use limited sensor nodes to improve network coverage performance, is the wireless sensor network in the application of harmful gas detection technical hotspots and difficulties.
Point 4: What are the attacks that can be mitigated by this proposed method?
Response 4:
Thank you very much for your valuable comments, the research direction of this article is to use the priority mechanism to patch the algorithm, improve the network detection coverage within the detection range, reduce node redundancy, and do not study the direction of network attacks, in future research, we will continue to study the content of network attack directions.
Point 5: What is the session time or lifetime of the randomly generated static nodes? Are they permanent nodes or temporary nodes?
Response 5:
Randomly generated static nodes have a limited lifetime and are temporary nodes. When the node energy is exhausted and the lifetime ends, a coverage hole is generated, and the mobile sensor node is used to repair the hole and extend the lifetime of the wireless sensor network.
Point 6: How routing is performed among the randomly assigned static nodes in WSN?
Response 6:
Thank you very much for your valuable opinion, the paper explains that the length of the communication radius Rc between nodes is twice the sensing radius Rs, ensuring that sensors can communicate with each other, and the communication radius of the node can make a hop node exchange information with each other and perform routing. Previously, because my discussion was not clear enough, it has been restated in the text.
Point 7: Mentioned Win10 and Matlab2016b were used here to get the simulation results. Needs to elaborate on the implementation specification of the proposed work.
Response 7:
- In the MATLAB simulation software, the rectangular factory building of 100m*125m is first simulated;
- The detection range of the sensor is regarded as a circle, and 25 static nodes are randomly deployed, that is, 25 circles with a radius of 12m or 8m are generated in a rectangle to calculate the coverage before patching.
- Divide every three circles into a group, and use the covering hole positioning algorithm based on the outer circle to calculate the area size of the outer circle
- Determine the repair priority according to the area size of the outer circumference
- Make a circle with a radius of 8m at the center of the outer circumference, that is, a mobile repair node.
- After generating 15 mobile nodes, the patching is completed, and the coverage and redundancy after the patching are calculated.
Point 8: How many nodes authors have considered while in implementation?
Response 8:
In the implementation process, the plant area considered is 100m*125m, 25 static sensor nodes are initially deployed with detection radii of 12m and 8m, and 15 mobile sensor nodes for patching have a detection radius of 8m. Basically covers the entire plant and meets the testing requirements.
Point 9: Authors mentioned the proposed algorithm compared with VORCP and VORPH. Add a table to show the comparison results with the existing systems.
Response 9:
Thank you very much for your valuable comments, a comparison table with existing algorithms has been added to the article
|
|
Coverage |
Increased coverage |
Redundancy |
|
VORCP |
75.04% |
12.58% |
10.54% |
|
VORPH |
76.78% |
14.32% |
11.13% |
|
Our Algorithm |
79.13% |
16.67% |
8.24% |
Table 1. Algorithm comparison
Point 10: Suggested consider the following works on WSN
- 10.1109/WiSPNET51692.2021.9419459
- https://doi.org/10.1007/978-81-322-2126-5_18
- https://doi.org/10.1155/2022/9134036
Response 10:
They have been referred to and cited.

Round 2
Reviewer 3 Report
The authors updated the manuscript as per the suggested comments.
Recommend to further process.